# Effect of Antibiotic Exposure on *Staphylococcus epidermidis* Responsible for Catheter-Related Bacteremia

**DOI:** 10.3390/ijms24021547

**Published:** 2023-01-12

**Authors:** Cassandra Pouget, Clotilde Chatre, Jean-Philippe Lavigne, Alix Pantel, Jacques Reynes, Catherine Dunyach-Remy

**Affiliations:** 1Department of Microbiology and Hospital Hygiene, Bacterial Virulence and Chronic Infections, INSERM U1047, CHU Nîmes Univiversity Montpellier, CEDEX 09, 30029 Nîmes, France; 2Department of Infectious and Tropical Diseases, CH Perpignan, 66000 Perpignan, France; 3Department of Infectious and Tropical Diseases, IRD UMI 233, INSERM U1175, CHU Montpellier, University Montpellier, CEDEX 5, 34295 Montpellier, France

**Keywords:** adaptation, biofilm, ceftobiprole, daptomycin, exposure, linezolid, *Staphylococcus epidermidis*, vancomycin, virulence, whole-genome sequencing

## Abstract

Coagulase-negative staphylococci (CoNS) and especially *Staphylococcus epidermidis* are responsible for health care infections, notably in the presence of foreign material (e.g., venous or central-line catheters). Catheter-related bacteremia (CRB) increases health care costs and mortality. The aim of our study was to evaluate the impact of 15 days of antibiotic exposure (ceftobiprole, daptomycin, linezolid and vancomycin) at sub-inhibitory concentration on the resistance, fitness and genome evolution of 36 clinical strains of *S. epidermidis* responsible for CRB. Resistance was evaluated by antibiogram, the ability to adapt metabolism by the Biofilm Ring test^®^ and the in vivo nematode virulence model. The impact of antibiotic exposure was determined by whole-genome sequencing (WGS) and biofilm formation experiments. We observed that *S. epidermidis* strains presented a wide variety of virulence potential and biofilm formation. After antibiotic exposure, *S. epidermidis* strains adapted their fitness with an increase in biofilm formation. Antibiotic exposure also affected genes involved in resistance and was responsible for cross-resistance between vancomycin, daptomycin and ceftobiprole. Our data confirmed that antibiotic exposure modified bacterial pathogenicity and the emergence of resistant bacteria.

## 1. Introduction

Coagulase-negative staphylococci (CoNS) are common bacterial colonizers of the skin and mucous membranes in humans [1]. *Staphylococcus epidermidis* is the most frequently isolated species from human epithelial microbiota [2]. *S. epidermidis* is a commensal bacterium for the host, but in immunocompromised patients, it can be opportunistic and cause severe infections [3]. Health-care-associated infections (HAIs) are a significant cause of morbidity and mortality worldwide and represent an increasing problem in modern medicine [4]. Around four million patients are affected by HAIs in Europe annually [5], and the prevalence of nosocomial infections varies around 12% [6]. Patients admitted to intensive care units are particularly susceptible to these infections, not only due to their immunocompromised status, but also due to the use of invasive procedures and devices (e.g., catheters) [7]. Among the main causative agents, *S. epidermidis* represents 30% of catheter-related bacteremia (CRB) [8,9]. The infection caused by *S. epidermidis* is directly linked to its ability to form a biofilm [3,10] and to increase its antibiotic tolerance [11].

Colonization of percutaneous or implanted medical devices allows bacterial persistence through biofilm formation. Biofilm formation is a multi-step process in which the bacteria first adhere to the surface to be colonized, and subsequently accumulate into a multilayered cell structure [12,13]. Biofilm formation on indwelling medical devices can lead to serious, recalcitrant infections. Biofilm can be found on a variety of indwelling medical devices, such as prosthetic heart valves, central venous catheters, urinary catheters, contact lenses, etc. [14,15]. All biofilms share fundamental characteristics, i.e., the cells within the biofilm are protected by the extracellular matrix they produce, and this protective material can decrease the effectiveness of both antibacterial molecules and host defense mechanisms. Of biofilms found in clinical infections, more than 65% were related to those on indwelling devices [16,17]. As a consequence, the mortality rate of patients undergoing catheterization was considerable. *S. epidermidis* is a common cause of biofilm-mediated catheter device-related infection, since it possesses several virulence factors involved in this mechanism (e.g., *atlE, fbe, embp* and *ica* operons) [3,18,19,20].

Moreover, this species is frequently resistant to several antibiotic classes due to its exposure to antibiotics [3,21]. Indeed, this exposure helps increase the resistance by selective pressure, enhancing the prevalence of multidrug resistance that has gradually developed in recent years [22]. The impact of the high antibiotic consumption on virulence and the ability of strains to form biofilm has been little described for CoNS and notably for *S. epidermidis*.

Here, we studied a collection of clinical *S. epidermidis* strains isolated from CRB and evaluated the impact of a 15-day antibiotic exposure at sub-inhibitory concentration on their resistance, fitness and genome.

## 2. Results

### 2.1. Potential Virulence of S. epidermidis Isolated from Catheter-Related Bacteremia

Thirty-six *S. epidermidis* specimens isolated from CRB (and particularly implanted port catheters (12, 33%)) from the University Hospital of Montpellier were included in this study (Appendix A). The resistance profile showed that all isolates were resistant to penicillin (36, 100%), 83% were resistant to oxacillin (30), 69% to fusidic acid (25) and 64% to erythromycin and ofloxacin (both 23) (Appendix A).

#### 2.1.1. Evaluation of Virulence in an In Vivo Caenorhabditis Elegans Model

The virulence of the studied strains was evaluated using a nematode in vivo model and compared to the reference *S. epidermidis* ATCC 35984. A non-virulent *E. coli* isolate, OP50, was used as the nutrient for *C. elegans*. This OP50 condition was used as a negative control for infection.

All *S. epidermidis* strains killed the L4 stage worms significantly faster than the control OP50 (*p* < 0.001). The lethal time 50% (LT50) values were used to measure the virulence potential (VP) of studied strains with the formula VP = [1/(LT50 sample/LT50 reference strain)]. A strain with a VP < 1 was considered as a low virulent isolate, whereas VP > 1 was considered virulent. Twenty *S. epidermidis* isolates showed low virulence (VP < 1) compared to the virulent ATCC35984 (*p* < 0.01) (Figure 1). In contrast, 16 clinical *S. epidermidis* strains presented high virulence against the nematode model (VP > 1).

#### 2.1.2. Evaluation of Biofilm Formation Ability

The ability of *S. epidermidis* to adhere and initiate biofilm formation was assessed by Biofilm Ring Test^®^ (Biofilm Control, Saint-Beauzire, France). The 36 clinical strains were compared to the reference ATCC35984, known to be a biofilm producer [23]. The values of the Biofilm Formation Index (BFI) were used to measure the biofilm formation potential (BFP) of studied strains using the formula BFP = [1/(BFI sample/BFI reference strain)]. A strain with BFP < 1 was considered to have a low potential of biofilm formation, whereas BFP > 1 presented a high ability to form biofilm. After 4 h, 28 clinical *S. epidermidis* strains showed a low potential to form early biofilm compared to ATCC35984, of which only 12 strains showed a significant difference (*p* < 0.05) (Figure 2). Conversely, eight strains presented a high ability to form biofilm, with half of these strains showing a significant difference (*p* < 0.01).

### 2.2. Effects of Exposure to Sub-Inhibitory Concentrations of Antibiotics on Virulence and Resistance of S. epidermidis

To evaluate the impact of antibiotic exposure on *S. epidermidis*, we selected five strains with varied virulence and potential for biofilm formation profiles: strains 97 and 105 among the low VP group and strains 5, 26 and 82 among the high VP group. Moreover, strains 5, 82 and 105 were weaker biofilm producers (*p* < 0.01), and 26 and 97 were described as strong biofilm producers (*p* < 0.01).

The Minimum Inhibitory Concentrations (MICs) of four anti-Gram-positive antibiotics (vancomycin, linezolid, ceftobiprole and daptomycin) against those five initial strains (D0) were determined. All the strains were exposed to the four different antibiotics for 15 days to determine differences in characteristics (resistance profile, biofilm formation and genome modifications). The characteristics were compared for each strain before (D0) and after exposure to 15 days of sub-inhibitory antibiotic concentrations (0.5 × MIC). A control of each strain after culture for 15 days without antibiotic was made.

#### 2.2.1. Effect on Antibiotic Susceptibility of *S. epidermidis*

All five studied clinical *S. epidermidis* strains were susceptible to the four antibiotics tested at D0 (Table 1). After 15 days of antibiotic exposure, all strains were resistant to vancomycin according to CA-SFM recommendation’s (Antibiogram Committee of the French Microbiology Society) (MICs = 4 mg/L), three out of five to daptomycin (MICs = 2 mg/L) and two out of five to ceftobiprole (MICs = 4 mg/L). In contrast, linezolid exposure increased the MICs of the strains, but none were resistant to this antibiotic after 15 days.

All *S. epidermidis* strains with increased vancomycin MIC after antibiotic exposure harbored a resistance to daptomycin (both MIC = 4 mg/L), whereas these mutants were susceptible to linezolid and ceftobiprole (Appendix A). Only one daptomycin-resistant strain (82D) was co-resistant to vancomycin. No other co-resistance was observed.

Interestingly, all resistant strains obtained after 15 days of antibiotic exposure retained this phenotype after a washout period of 15 days (Table 1).

#### 2.2.2. Effect on Biofilm Formation of *S. epidermidis*

We focused our study on the strains that became resistant to antibiotics after the exposure. Thus, the change in biofilm formation after antibiotic exposure was assessed by Biofilm Ring Test^®^ to compare the selected strains (5V, 26V, 82V, 97V, 105V, 82D, 97D, 105D, 82C and 105C) and the initial ones (5, 26, 82, 97 and 105). A control of 15 days of culture without antibiotics for each strain was made with results similar to the initial strains at D0.

After a 4 h incubation, all exposed strains were able to form early biofilm (BFP > 1). In all cases, the resistant *S. epidermidis* strains demonstrated a significant increase in their ability to form biofilm with BFP > 1 even for strains 26 and 97, which were stronger biofilm producers at the initial stage. After vancomycin exposure, we observed: 5 vs. 5V, BFP = 0.23 ± 0.01 vs. 2.3 ± 0.2 (*p* < 0.01); 26 vs. 26V, 2.8 ± 0.07 vs. 3.1 ±0.3 (*p* < 0.05); 82 vs. 82V, 0.41 ± 0.03 vs. 3.6 ± 0.2 (*p* < 0.001); 97 vs. 97V, 3.1 ± 0.03 vs. 3.4 ± 0.2 (*p* < 0.1) and 105 vs. 105V, 0.35 ± 0.02 vs. 2.09 ± 0.1 (*p* < 0.01) (Figure 3). After daptomycin exposure, we also noted 105 vs. 105D, BFP = 0.35 ± 0.02 vs. 3.9 ± 0.2 (*p* < 0.001); 82 vs. 82D, 0.41 ± 0.03 vs. 4.3 ± 0.1 (*p* < 0.001) and 97 vs. 97D, 3.1 ± 0.03 vs. 4.2 ± 0.3 (*p* < 0.001) (Figure 3). Interestingly, no significant effect was noted for the two strains exposed to ceftobiprole, although their BFP profile had changed: 82C, 0.41 ± 0.03 vs. 1.8 ± 0.1 (*p* = not significant (ns) and 105 vs. 105C, 0.35 ± 0.02 vs. 1.2 ± 0.09 (*p* = ns)) (Figure 3).

#### 2.2.3. Effect of Antibiotics on Their Ability to Inhibit *S. epidermidis* Biofilm Formation

Using the Antibiofilmogram^®^, we determined the partial MIC biofilm (MICb) of each strain after a 4 h incubation (Figure 4). Vancomycin, ceftobiprole and daptomycin were significantly less able to inhibit the biofilm formation for all strains tested with partial MICb higher than the MIC (*p* < 0.001) (Figure 4A,C,D). Linezolid inhibited the biofilm formation at a similar concentration to the MICs for all vancomycin-resistant strains (5V, 26V, 82V, 97V and 105V) (*p* = ns), whereas this antibiotic was significantly less able to inhibit biofilm formation for daptomycin- and ceftobiprole-resistant strains (*p* < 0.001) (Figure 4B).

#### 2.2.4. Effect on the Genomes of *S. epidermidis* after Antibiotic Exposure

The 15 *S. epidermidis* strains developing resistance (initial (D0); resistant after antibiotic exposure (D15) and after antibiotic washout (D30)) were sequenced to determine the change between D0, D15 and D30 and to compare the genome content of all strains (Table 2). Firstly, analysis confirmed that all pairs of strains belonged to the same sequence type (ST) before and after antibiotic exposure. ST35 was detected in two of five *S. epidermidis* pairs and ST640, ST22 and ST87 in the remaining pairs (Table 2).

All strains harbored the *mecA* gene at D0, confirming the antibiogram results. Moreover, the genomes contained a mutation in *blaZ* (encoding penicillin resistance), *dfrC* (encoding trimethoprim resistance), *norA* (corresponding to fluoroquinolone resistance), *fusA* (corresponding to p.L461S conferring acid fusidic resistance), *msrA* (encoding for a macrolide efflux protein) and *fosB* (conferring a fosfomycin resistance). Some differences could be noted. The genomes of strains 5 and 97 also shared the presence of *tetK*, *aadD*, *mphC* and *ant(4″)-*Ib genes that encode a tetracycline efflux pump protein, an aminoglycoside adenylyltransferase, a macrolide phosphotransferase and a kanamycin nucleotidyltransferase, respectively. Moreover, strains 82 and 105 harbored *aph(2″)*-Ia and *qacA* (mediating resistance to quaternary ammonium compounds) genes. Interestingly, *mgrA* (a regulator of several efflux pumps implicated in tetracycline and quinolone resistances) was only detected in one strain (97). Finally, *ant(9)-*Ia (conferring aminoglycoside resistance), *vatB* and *vgaA* (both conferring resistance to streptogramin A and related compounds) and the *ermA* gene (conferring resistance to erythromycin and clindamycin) were only detected in strain 105 (Table 2).

Concerning the virulome, all *S. epidermidis* strains harbored a similar panel of known virulence-factor-encoding genes, notably the presence of the *ica* gene that encodes N-acetylglucosaminyltransferase, the enzyme involved in PIA synthesis, and multiple copies of the insertion sequence *IS256* (Table 2).

Moreover, each pair of strains was considered identical based on the low number of SNP differences (<71 SNP) and according to Ankrum and Hall criteria [25] (Table 3). SNPs concerned different mutations affecting genes classified according to their functions. Few mutations were detected in functional genes, mainly affecting antibiotic-resistance-encoding genes (Table 3). D15 and D30 were identical.

## 3. Discussion

The pathogenicity of CoNS is particularly diverse and varied between species. Among them, *S. epidermidis* is part of the normal human cutaneous microbiota that becomes an important opportunistic pathogen causing HAIs and tends to be multidrug-resistant [3]. The virulence of *S. epidermidis* and notably its ability to form biofilm are important factors allowing protection against antibiotic action and the host defense immune system [26]. These bacteria can modulate their pathogenicity due to the influence of different factors, such as exposure to varying concentrations of antibiotics [27,28]. Recently, bacterial virulence and adhesion to biomaterials have gained increased attention [29]. Implanted medical devices may facilitate infection, since any *S. epidermidis* strain inadvertently introduced into a surgical site is capable of rapidly adhering to the surface of the device. This surface-associated bacterial growth is known as biofilm formation and appears to be the key factor enabling invasive CRB for an otherwise largely non-pathogenic microorganism. The ubiquitous presence of *S. epidermidis* on human skin has enabled infection to emerge as a major complication of medical devices [3,30].

In this study, we observed the difference in virulence and biofilm formation profiles of a collection of 36 clinical *S. epidermidis* strains, with the majority possessing a low virulence potential in a *C. elegans* model and a low ability to form biofilm. However, some strains had a high virulence profile, confirming that *S. epidermidis* is not always an “accidental pathogen” as previously described [2]. Several known virulence genes of *S. epidermidis* are shared by commensal and pathogenic strains [2,3]. Among them, we found the phenol-soluble modulins (PSMs), involved in inflammatory response and lysis of leukocytes that were present in the genomes of all *S. epidermidis* strains of this study [31]. The ESAT-6 secretion system (ESS) (*esaAB, esaABC* and *esaAC* genes) implicated in immune system evasion and neutrophil elimination was also present in four of the five sequenced strains [32,33]. Cytotoxins and hemolysins were also identified, but although these toxins are important in the pathogenesis of *S. aureus*, their role in *S. epidermidis* infections is unknown. Few reports have described the presence of cytotoxin-encoding genes and their expression in CoNS [34], such as *hla* and *hld* genes (that encode α- and δ-hemolysin, respectively) that have been identified in *S. epidermidis* [35]. α-Hemolysin exerts hemolytic, dermonecrotic and neurotoxic effects [36], while β-toxin possesses phosphorylase activity and high affinity for the cell membrane of different types of cells, causing membrane instability [37]. Recently, Pinheiro et al. observed a high frequency of the *hld* gene in nosocomial isolates with high toxigenic potential. This suggests a potential role of these cytotoxin genes in the establishment of the pathogenicity of *S. epidermidis*. Further experiments are needed to clearly understand all the factors involved in *S. epidermidis* virulence, but these findings correlate with our data (91% of our sequenced strains were positive for *hld*). Moreover, our data demonstrate an important role of these cytotoxin genes in the establishment of these species and possibly in the development of infections caused by CoNS [38].

One of our main objectives was to understand the impact of antibiotics on the genomes of *S. epidermidis* and their consequences on resistance profile and biofilm formation. Commensal bacteria are exposed to selective pressure due to the high consumption of antibiotics followed by a high resistance level of species against a variety of antibiotics, notably in bacteremia [39]. However, very few studies have evaluated this impact. Here, we showed that antibiotic pressure selected resistant or reduced-sensitivity *S. epidermidis* isolates using vancomycin, daptomycin and ceftobiprole. Interestingly, linezolid exposure did not lead to resistance, although two *S. epidermidis* strains increased their MICs to the limit of susceptibility (strains 26 and 105, both MIC = 4 mg/L). Moreover, for the three other antibiotics, although vancomycin induced resistance in all studied strains, daptomycin and ceftobiprole had a more mitigated effect (two and three strains were resistant after antibiotic exposure, respectively). Vancomycin resistance was acquired after a short exposure (between 2 and 10 days), as previously observed [40]. We confirmed previous studies showing that *S. epidermidis* strains rapidly developed a stable resistance to vancomycin when subcultured with sub-inhibitory concentrations of vancomycin, and without reversion when subcultured without vancomycin. Indeed, the five strains did not reverse their MICs after 15 days without this antibiotic. This acquisition of vancomycin resistance was associated with an increased cell wall thickness in *S. aureus* and *S. capitis* [41,42]. However, the molecular determinants of this vancomycin resistance remain unknown. Some authors suggest a role for the *murA* gene in catalyzing the first step of peptidoglycan biosynthesis in *Staphylococcus* sp. [43]. Notably, the *murA* gene was mutated in four out of five sequenced strains in our study and could represent a target to be explored. Moreover, one strain, 5V, had a mutation in *rpoB* (A477D), which is also involved in decreased susceptibility of vancomycin [44]. Cross-resistance has already been described in *S. aureus* strains resistant to vancomycin, notably concerning the resistance to daptomycin without previous exposure [45,46]. This phenomenon has not yet been described in *S. epidermidis*. For *S. epidermidis* co-resistance, only one study has induced resistant isolates using the mutant prevention concentration and the mutant selection window and found a preferential co-resistance between β-lactam and tetracyclines and between β-lactam and aminoglycosides [47]. Our results showed that, after vancomycin exposure, the strains increased their MIC values generated in three out of five resistant strains. Acquisition of daptomycin resistance is a stepwise and multifactorial process including cell membrane and cell wall perturbations [48,49]. Mutations have been described in various genes. Interestingly, in our study, each of the three *S. epidermidis* strains resistant to daptomycin presented different mutations. One was in the *vraS* gene for strain 97, a gene of the two-component system VraSR that represents a key regulator of cell wall biosynthesis and plays an important role in *S. aureus*. This mutation (E276K) was previously described in *S. aureus* resistant to daptomycin [50]. Another publication highlighted a vancomycin resistance associated with the VraSR regulatory system that also modulates biofilm formation in *S. epidermidis* [51]. A second mutation was in the *dltA* gene for strain 105, a cell envelope charge gene previously described in daptomycin resistance [52]. Finally, a mutation was found in the *graR* gene for strain 82, a gene of the two-component system GraSR, which regulates the peptidoglycan synthesis pathway in *S. aureus* and was previously described in co-resistance to vancomycin and daptomycin [53,54], as confirmed in our study. We also observed two ceftobiprole-resistant strains with a mutation C197Y in PBP2 (strains 82C and 105C). This mutation has been described in clinical methicillin-resistant *S. aureus* [55]. No co-resistance was noted in the two mutated strains. Finally, alternatives to vancomycin must be considered in the case of *S. epidermidis* CRB. Vancomycin resistance was associated with an increase in daptomycin MICs, but not in linezolid MICs, confirming that this antibiotic appears as an excellent alternative. Moreover, our study highlighted that prolonged exposure to linezolid did not induce any intrinsic or cross-resistance.

We highlighted the adaptability of *S. epidermidis*, comprising reduced antibiotic susceptibility, modification of virulence and increased biofilm formation potential to circumvent the host immune response and the antibiotic treatment and to favor persistent infection, as observed for *S. aureus* [56]. In our study, strains exposed to daptomycin and vancomycin clearly increased their ability to form biofilm. This adaptation has been previously described in *S. epidermidis*, particularly after contact with vancomycin [57], macrolides [58] and fluoroquinolones [59]. However, in these cases, exposure was of short duration. Rachid et al. noted that *S. epidermidis* increased the expression of *ica* and regulated the *agr* system in the presence of a low concentration of macrolides [60]. This could explain the phenomenon observed in this study. *S. epidermidis* adaptation has been discussed in a recent publication in strains isolated from prosthetic joint infection, demonstrating the multifactorial processes of infection adaptation and how *S. epidermidis* flexibly repurposes and edits factors important for colonization to facilitate survival in hostile infection environments [61].

We also noted that the daptomycin-resistant strains remained high biofilm producers (Figure 3). However, using Antibiofilmogram^®^, these strains were particularly affected by the presence of linezolid, which significantly modified their ability to form biofilm, rendering them more resistant to this antibiotic (Figure 4). We also observed that ceftobiprole, a molecule never previously tested using this assay, was ineffective against the biofilm formation of *S. epidermidis*. However, these strains were sensitive to this molecule, confirming the need to develop new solutions to evaluate antibiotic activity on bacteria that could complement the “classical” antibiogram. Similarly, daptomycin, which had no activity on the biofilm formation at maximum concentrations of 32 mg/L in this study, was known to be effective at high doses to eliminate bacteria organized in biofilm [62,63]. According to our results, daptomycin or ceftobiprole do not seem to be the most suitable in vitro candidates to eradicate bacterial biofilm. Indeed, few studies present the effectiveness of ceftobiprole on CoNS and in particular on *S. epidermidis*. A publication by Henriksen et al. described this molecule as a good potential therapeutic treatment, since all the clinical strains isolated in this study had very low MICs [64]. However, no study in the literature mentions the potential antibiofilm effect of ceftobiprole on CoNS, despite its efficacy against *S. aureus* [65].

*S. epidermidis* biofilms contain a high number of persister cells upon antibiotic exposure [66]. We investigated this aspect by a genomic comparison of the strains before and after antibiotic pressure. After the 2-week exposure, we observed a very low genetic diversity between the different isolates before and after antibiotic exposure (Table 3). One to twenty-four SNPs were noted, confirming that the strains were the same according to the Ankrum and Hall criteria, which defined cut-offs classifying ≤71 SNPs as the “same”, 72–123 SNPs as “very closely related”, 124–156 SNPs as “closely related” and ≥157 SNPs as “distantly related” [25]. Moreover, no modification in genome size was observed between the two periods, suggesting that the bacterial adaptation in this hostile environment did not require the reduction or major modifications of the genome. The mutations mainly concerned genes involved in the resistance mechanisms observed. Interestingly, the vancomycin-resistant strain 5V also harbored a mutation in the *rpoB* gene, which encodes the beta subunit of the RNA polymerase (RNAP) and is the target of rifampicin. RpoB was previously shown to mutate quickly both in vitro and in vivo upon antibiotic exposure [67]. Mutations in specific positions are known to affect the drug binding to the RNAP, leading to drug co-resistance. While it is easy to predict which mutation caused resistance to antibiotics, the low number of SNPs found in our strains before/after antibiotic pressure more likely point to tolerance acquisition. Determining which mutations might have affected tolerance is very difficult. Transposon mutagenesis studies further indicated that a limited number of single genes affect antibiotic tolerance, such as toxin–antitoxin systems and the stringent response pathway [68]. However, the relevance of the stringent response in antibiotic tolerance is under debate for *S. aureus*, the closest relative of *S. epidermidis* [69,70]. Indeed, taken together these data suggest that the bacterial adaptation in this hostile environment does not require the reduction or major modification of the genome, as previously noted. Moreover, a mutation in the *icaR* gene, a negative regulator of *icaADBC* expression, was detected [71]. However, no major modifications were noted in the biofilm formation of this strain, which significantly increases its ability to form biofilm similarly to the other studied strains (which did not harbor the *icaR* mutation). Thus, the regulation of *ica* is a complex and multifactorial process, involving a variety of external environmental factors and internal regulators [72].

In summary, this study highlights the influence of antibiotic exposure on *S. epidermidis* by rapid genome modification affecting the resistance genes and by an evolution of the regulation of genes involved in biofilm, allowing this opportunistic bacterium to adapt to its environment.

## 4. Materials and Methods

### 4.1. Bacterial Strains, Culture Conditions and Antimicrobial Susceptibility Testing

All *S. epidermidis* strains belong to the collection of our Department of Microbiology at Nîmes University Hospital (France). They were isolated from clinical blood samples after diagnosis of CRB (≥3 isolates from peripheral blood cultures and 1 isolate from central blood culture with identical antibiograms) over one year. Thirty-six consecutive strains were included in this study, each strain isolated from a different patient. The reference strain of *S. epidermidis* ATCC 35984 was used as a control of biofilm formation and virulence potential because this strain was previously described as virulent [23]. *E. coli* OP50 was the nutrient of *Caenorhabditis elegans* and served as a negative control in the worm model. Bacterial identification was obtained by mass spectrometry (Vitek-MS^®^, Biomérieux, Marcy-l’Étoile, France).

Antimicrobial susceptibility testing of isolates was performed by disc diffusion test and broth microdilution method on Muller Hinton (MH, Bio-Rad, Marnes-La-Coquette, France) according to CA-SFM recommendations v 1.0 [24]. Vancomycin MICs were determined using broth microdilution procedures (UMIC) (Bruker, Champs-sur-Marne, France). MICs were also measured before and after antibiotic exposure.

Bacteria were subcultured in liquid media overnight at 37 °C under shaking conditions at 200 rpm or plated in Luria Broth (LB; Invitrogen, ThermoFisher, MA, USA).

### 4.2. Nematode Killing Assay

The nematode infection assay was carried out as previously described using the Fer-15 mutant line worms, fertile at 15 °C and sterile at 25 °C [73]. Fer-15 was provided by the *Caenorhabditis* Genetics Center. Nematodes were synchronized at the development stage L4 using hypochlorite method. Overnight cultures of the studied strains in LB were harvested, centrifuged and resuspended in phosphate buffered saline solution (PBS) at a concentration of 10^5^ CFU/mL. Nematode growth medium (NGM) agar plates were inoculated with approximatively 30 L4 stage worms and incubated at 25 °C. The worm survival rate was assessed daily with a stereomicroscope (Nikon SMZ 745, Amstelveen, The Netherlands). These experiments determined the lethal time 50% (LT50), which corresponded to time (in days) required to kill 50% of the initial worm population. The values of LT50 were used to measure the virulence potential (VP) using the formula VP = [1/(LT50 sample/LT50 reference strain)] for each strain. Strains with values of VP < 1 were considered to present a low virulence compared to ATCC35984, whereas strains with VP > 1 were considered to harbor a high virulence. All experiments were performed in biological triplicate, repeated twice for each selected strain.

### 4.3. Biofilm Formation

The BRT^®^ (Biofilm Control), which measures the mobility of superparamagnetic microbeads subjected to a magnetic field, was used to study the early stages of biofilm formation according to the manufacturer’s recommendations [74]. Briefly, bacterial cultures standardized at a concentration of 0.5 Mac Farland were incubated at 37 °C in 96-well microtiter plates in the presence of magnetic beads. At set time-points, the plates were placed on a magnetic block in the reader. The images of each well before and after magnetic attraction were analyzed with the BioFilm Control software (BFC Elements^®^ 3), which gave a Biofilm Formation Index (BFI). The adhesion ability of each strain was expressed as this BFI that is inversely proportional to the attached cell number. Two incubation times were defined: 2 and 4 h. After 4 h of incubation, BFI values discriminated strains with different biofilm behaviors. The BFI values were used to measure the biofilm formation potential (BFP) using the formula BFP = [1/(BFI sample/BFI reference strain)] for each strain. Strains with values of BFP < 1 were considered to be a weaker biofilm producer, whereas strains with BFP > 1 were considered to be a stronger biofilm producer compared to ATCC35984. All experiments were performed in biological triplicate and repeated twice for each selected strain.

### 4.4. Antibiotic Exposure

The selected strains were exposed for 15 days to sub-inhibitory concentration (0.5 × MIC) of several commonly used antibiotics (ceftobiprole, daptomycin, linezolid and vancomycin) followed by a 15-day washout period (Appendix A). Each day, bacteria were subcultured in liquid MH medium. Bacteria were diluted each day to optical density 0.1 using a spectrophotometer (Jenway 6320D, Fisher-Scientific, Waltham, MA, USA) for a total volume of 3 mL. Fresh antibiotic at sub-inhibitory concentration was added each day for 15 days total.

### 4.5. Antibiofilmogram^®^

Partial MICbs were measured using Antibiofilmogram^®^ test (Biofilm Control) according to the manufacturer’s recommendations [75]. Briefly, experiments were performed with the isolates before and after antibiotic exposure. The 96-well microtiter plates containing bacteria, magnetic beads and antibiotic solutions were incubated at 37 °C for 4 h before visual reading. At this time, the plates were placed onto a magnetic block, read after magnetic attraction for 1 min and analyzed using a microplate scanner with BioFilm Control software (BFC Elements 3.0), which generated a Biofilm Formation Index (BFI). A second algorithm was used to calculate the partial biofilm minimal inhibitory concentration (partial MICb), which represented the concentration at which the antibiotic inhibited the adhesion step of biofilm formation. This MICb was assessed for several antibiotics (ceftobiprole, daptomycin, linezolid and vancomycin). Four wells without antibiotics, filled with the bacterial suspension and magnetic beads, were used as the positive control. Assays were performed in triplicate.

### 4.6. Whole-Genome Sequencing

Genomic DNA (gDNA) was extracted from 3 mL of overnight cultures of the strains using DNeasy UltraClean Microbial Kit (Qiagen, Hilden, Germany) according to the manufacturer’s instructions. Quality of gDNA was examined using Qubit fluorometer 2.0 (Invitrogen, Waltham, MA, USA). Fifteen *S. epidermidis* strains were sequenced using Illumina MiSeq Sequencing system (Illumina, San Diego, CA, USA) with Nextera XT DNA Library Prep Kit (paired-end read libraries, Illumina) according to supplier’s recommendations. Quality control of the reads was performed with FastQC software (v.0.11.7). CLC Genomics Workbench software (Qiagen, Germantown, MA, USA) was used for genome assemblies, bacterial genome annotation and MLST. ResFinder 4.1, VirulenceFinder 2.0 and PlasmidFinder 2.1 were used for sequence analysis [76,77,78,79]. Single-nucleotide polymorphisms (SNPs) were called using Snippy [80]. SNP numbers between parental and antibiotic-exposed strains were interpreted according to the criteria of Ankrum and Hall [25], which defined strains with ≤71 SNPs as the “same” strains, strains with 72–123 SNPs as “very closely related” strains, strains with 124–156 SNPs as “closely related” strains and strains with ≥157 SNPs as “distantly related” strains.

### 4.7. Statistical Analysis

Statistical analyses were performed using GraphPad Prism version 9.2. Tests used for the *p*-value determination are mentioned in figure legends. For the nematode killing assays, differences in survival rates between strains were tested by a log-rank (Mantel–Cox) test for statistical significance. The biofilm formation was compared by Mann–Whitney test.

## 5. Conclusions

Antibiotic exposure of *S. epidermidis* strains had a direct impact on their pathogenicity. Prolonged culture in sub-inhibitory concentrations of vancomycin, ceftobiprole and daptomycin increased the abilities of these clinical bacteria to form biofilm and to develop cross-resistances to other antibiotics due to rapid genomic mutations. In contrast, linezolid had no clear influence on the resistance profile and the biofilm formation potential of strains. Better understanding of the mechanisms and processes of *S. epidermidis* adaptation under antibiotic pressure will help us to adapt more effective strategies against CoNS biofilm-related infection.

## Figures and Tables

**Figure 1 ijms-24-01547-f001:**
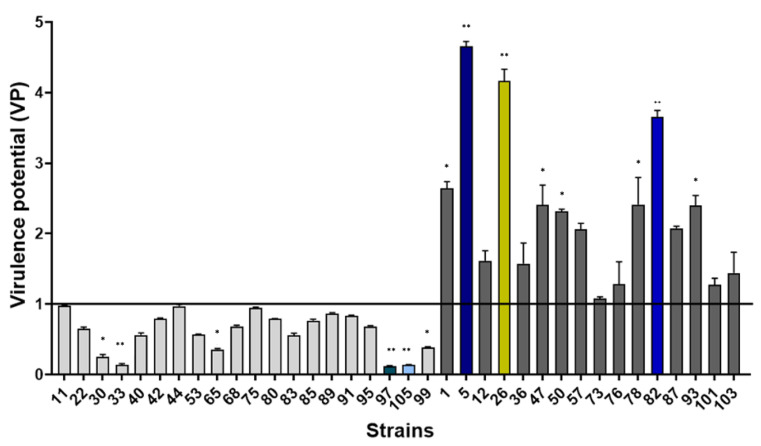
Virulence potential (VP) of 36 *S. epidermidis* clinical strains isolated from catheter-related bacteremia evaluated using an in vivo *C. elegans* model and compared to the reference virulent strain ATCC35984. Strains with values of VP < 1 were considered as low virulence, whereas strains with VP > 1 were considered as high virulence compared to ATCC35984. All experiments were performed in biological triplicate, repeated twice. Means ± standard errors are presented. Strains in color correspond to strains selected for the next experiments. Statistical differences between each clinical strain and the ATCC35984 were obtained using two-way ANOVA: *, *p* < 0.05; **, *p* < 0.01.

**Figure 2 ijms-24-01547-f002:**
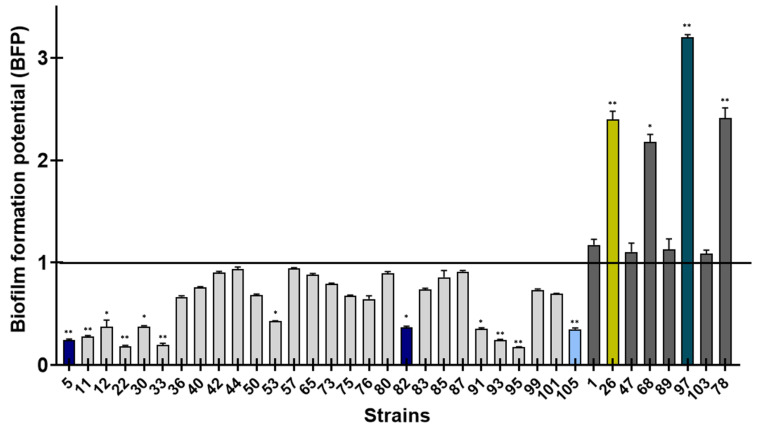
Biofilm formation potential (BFP) of 36 *S. epidermidis* clinical strains isolated from catheter-related bacteremia evaluated using the BioFilm Ring Test^®^ and compared to the reference strain ATCC35984 after 4 h incubation. Strains with values of BFP < 1 were considered to be weaker biofilm producers, whereas strains with BFP > 1 were considered to be stronger biofilm producers compared to the ATCC35984. All experiments were performed in biological triplicate, repeated twice. Means ± standard errors are presented. Strains in color correspond to strains selected for the next experiments. Statistical differences between each clinical strain and the reference ATCC were obtained using two-way ANOVA. *, *p* < 0.05; **, *p* < 0.01.

**Figure 3 ijms-24-01547-f003:**
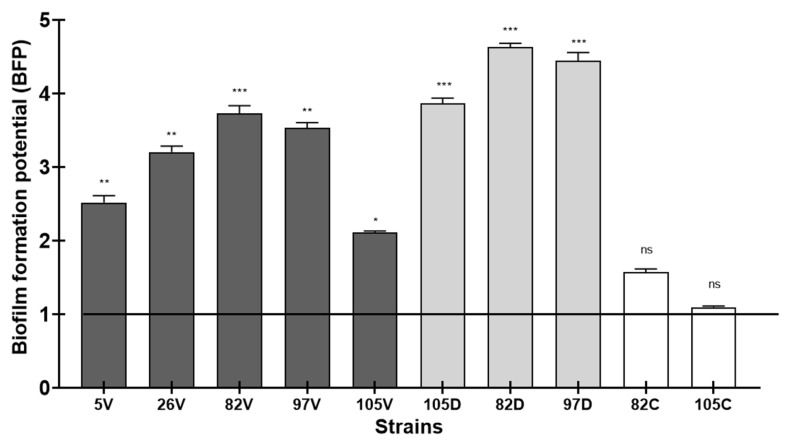
Biofilm formation potential of *S. epidermidis* strains isolated from catheter-related bacteremia and exposed to sub-inhibitory concentrations (0.5 × MIC) of different antibiotics (V: vancomycin (dark gray); D: daptomycin (light gray), C: ceftobiprole (white)) evaluated using the BioFilm Ring Test^®^ and compared to the initial strains after 4 h incubation. Strains with values of BFP < 1 were considered to be weaker biofilm producers, whereas strains with BFP > 1 were considered to be stronger biofilm producers compared to the initial bacteria. All experiments were performed in biological triplicate, repeated twice. Means ± standard errors are presented. Statistical differences between pairs of clinical strain at D0 and D15 were obtained using two-way ANOVA: *, *p* < 0.05; **, *p* < 0.01; ***, *p* < 0.001.

**Figure 4 ijms-24-01547-f004:**
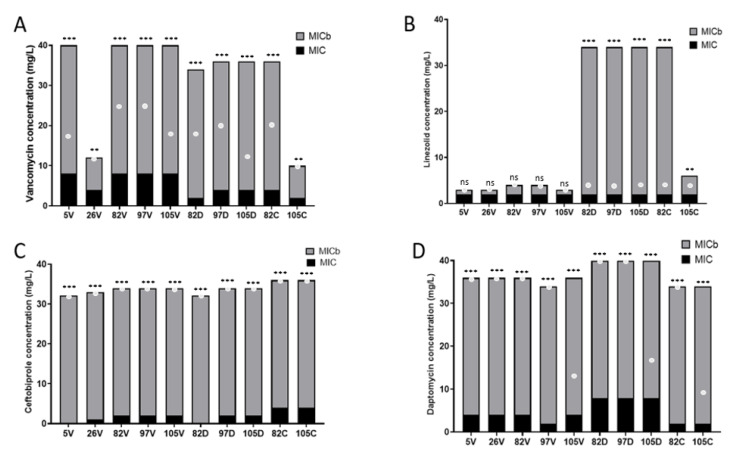
Comparison of MICs and partial MICbs of *S. epidermidis* clinical strains isolated from catheter-related bacteremia and exposed to sub-inhibitory concentrations (0.5 × MIC) of different antibiotics (V: vancomycin, D: daptomycin and C: ceftobiprole) and evaluated using the Antibiofilmogram^®^ assay. The MICs and MICbs were measured in contact with vancomycin (**A**), linezolid (**B**), ceftobiprole (**C**) and daptomycin (**D**). The white dots represent the values of the MICb obtained before antibiotic exposure at D0. All experiments were performed in biological triplicate. Means ± standard errors are presented. Statistical differences between pairs of MICs and MICbs after antibiotic exposure were obtained using two-way ANOVA: **, *p* < 0.01; ***, *p* < 0.001.

**Table 1 ijms-24-01547-t001:** Change in MICs before and after 15 days of antibiotic exposure and after 15 days without antibiotics for the five selected *S. epidermidis* strains. MICs were compared before/after exposure to 0.5 × MIC for 15 days using vancomycin, daptomycin, linezolid and ceftobiprole. MICs in bold represent strains that became resistant. All experiments were performed in biological triplicate.

	Vancomycin MIC (mg/L)			Daptomycin MIC (mg/L)			Linezolid MIC (mg/L)			Ceftobiprole MIC (mg/L)		
Strain	Before (D0)	After ATB (D15)	After No ATB (D30)	Before (D0)	After ATB (D15)	After No ATB (D30)	Before (D0)	After (D15)	After No ATB (D30)	Before (D0)	After ATB (D15)	After No ATB (D30)
5	1 (S) *	**4 (R)**	**4 (R)**	0.5 (S)	0.5 (S)	1 (S)	1 (S)	1 (S)	1 (S)	0.25 (S)	1 (S)	1 (S)
26	1 (S)	**4 (R)**	**4 (R)**	1 (S)	1 (S)	1 (S)	1 (S)	4 (S)	4 (S)	0.5 (S)	1 (S)	2 (S)
82	1 (S)	**4 (R)**	**4 (R)**	0.5 (S)	**2 (R)**	**2 (R)**	1 (S)	2 (S)	2 (S)	1 (S)	**4 (R)**	**8 (R)**
97	1 (S)	**4 (R)**	**8 (R)**	0.5 (S)	**2 (R)**	**4 (R)**	1 (S)	2 (S)	4 (S)	0.5 (S)	1 (S)	1 (S)
105	1 (S)	**4 (R)**	**4 (R)**	0.5 (S)	**2 (R)**	**2 (R)**	1 (S)	4 (S)	2 (S)	0.5 (S)	**4 (R)**	**4(R)**

* S, susceptible; R, resistant according to CA-SFM recommendations [24].

**Table 2 ijms-24-01547-t002:** Genomic features, virulome and resistome of *S. epidermidis* strains isolated from catheter-related bacteremia.

		Strain 5	Strain 26	Strain 82	Strain 97	Strain 105
Genotyping	Genome Size (bp)	2,517,447	2,509,906	2,553,400	2,475,519	2,528,716
	No. of contigs	23	21	56	41	40
	No. of ORFs *	2262	2264	2309	2242	2267
	Sequence type	35	640	22	35	87
Virulome		*agr, hld, psmα, icaA,B, IS256, bhp, atlE* and *esaAB*	*agr, hld, psmα, icaAB, IS256, bhp, atlE* and *esaAB*	*agr, hld, psmα, psm mec, icaAB, IS256, bhp* and *atlE*	*agr, hld, psmα, icaA,B, IS256, bhp, atlE* and *esaAB*	*agr, hld, psmα, psm mec, IS256, bhp, atlE* and *esaAB*
Resistome		*blaZ, mecA, dfrC, norA, fusB, tetK, fosB, msrA, aad, ant(4″)*-Ib and *mphC*	*mecA, msrA, fusB, dfrC* and *norA, fosB*	*blaZ, mecA, aph(2″)*-Ia*, fusB, dfrC, norA, qacA* and *fosB*	*blaZ, mecA, mphC, ant(4″)-*Ib*, aad, msrA, fusB, dfrC, tetK, morA, mgrA* and *fosB*	*blaZ, mecA, ant(9)-*Ia, *aph(2″)-*Ia, *vatB, ermA, fusB, vgaA, dfrC, norA, fosB* and *qacA*

* ORF, open reading frame.

**Table 3 ijms-24-01547-t003:** Number of SNPs detected and mutations directly affecting virulence- or resistance-encoding genes for each *S. epidermidis* strain after exposure to sub-inhibitory concentration of different antibiotics.

Number of SNP	Strain 5	Strain 26	Strain 82	Strain 97	Strain 105
After vancomycin exposure	24	6	3	11	2
After ceftobiprole exposure	-	-	1	-	4
After daptomycin exposure	-	-	2	3	3
Localization of SNP concerning known genes	*rpoB, IS256* and *icaR*	*murA*	*murA* (vancomycin), *pbp2* (ceftobiprole) and *graR* (daptomycin)	*murA, rpoB* (vancomycin) and *vraS* (daptomycin)	*murA* (vancomycin), *pbp2* (ceftobiprole) and *dltA* (daptomycin)

## Data Availability

Not applicable.

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
