# Peer review of "Effect of Antibiotic Exposure on Staphylococcus epidermidis Responsible for Catheter-Related Bacteremia"

_ijms, 2023, doi:10.3390/ijms24021547_

Round 1

Reviewer 1 Report

Effect of antibiotics exposure on Staphylococcus epidermidis responsible for catheter-related bacteremia by Cassandra Pouget et al

The manuscript covers a very interesting and important subject that’s has potential ramifications across the healthcare systems globally. S. epidermidis is currently the most frequently cause of central line-associated bloodstream infections and the second-most common organism isolated from hip and knee prosthetic joint infections. During several years spread of several endemic multidrug-resistant S. epidermidis strains as occurred and they predominate now across healthcare systems globally limiting the treatment options for indwelling and prosthetic device infections that are already difficult to treat. So new strategies for optimal management of S. epidermidis infections are urgently needed.

Even though the topic is interesting I do have a few comments that the authors should address in order to increase the understanding and clinical ramifications of the manuscript.

Major comments:

Abstract

Line 28-30: Our data confirmed that antibiotic exposure modified bacterial pathogenicity and, in this situation, linezolid seems to be the best adapted therapeutic option to limit the emergence of resistant bacteria

Spread of linezolid-resistant S. epidermidis isolates (cfr-carrying plasmid) is well documented in the hospital setting, also with interspecies transfer to S. aureus. https://pubmed.ncbi.nlm.nih.gov/35425984/ In light of this, the assessment of linezolid superiority should be limited to the results of the current study

Results

Line 63-64: is it possible to in specify the period of time that the isolates were obtained? I would also be interesting, and maybe also important for the interpretation of the results, any antimicrobial treatment given to the cases prior/during the time of detection of isolates in blood culture. And also confirm if the S. epidermidis isolates were obtained from 36 different patients.

Line 121-122: After 15 days of antibiotics exposure, all strains were resistant to vancomycin (MICs= 4mg/L), 3/5 to daptomycin (MICs= 2mg/L) and 2/5 to ceftobiprole (MICs= 122 4 mg/L).

According to EUCAST Clinical Breakpoint Tables v. 12.0, valid from 2022-01-01, Vancomycin MIC breakpoints for coagulase-negative staphylococci is S ≤ 4 mg/L and R > 4 mg/L. The authors interpret the breakpoints as if R 4 mg/ml for Vancomycin. This discrepancy has to be addressed as it has major implications for the interpretation of the results. In addition, ceftaroline MICs for coagulase-negative staphylococci is missing. Please comment.

Line 127: * S, susceptible; R, resistant according to EUCAST recommendations [27].

This may be wrong? Reference 27 is currently Dai, Y.; Wang, Y.; Liu, Q.; Gao, Q.; Lu, H.; Meng, H.; Qin, J.; Hu, M.; Li, M. A novel ESAT-6 secretion system-secreted protein 518 EsxX of community-Associated Staphylococcus aureus lineage ST398 contributes to immune evasion and virulence. Front Micro-519 biol 2017, 8, 819.

Materials and methods

Line 353-355: Antimicrobial susceptibility testing of isolates was performed by disc diffusion test 353 and broth microdilution method on Muller Hinton according to EUCAST recommendations. Please include which version (v 12.0?)

Line 181-184: It would be interesting to include Table 2 WGS data for the 15 sequenced strains. I suppose that this refers to train 5, 26, 82, 97 and 105 at D0, D15 and D30? Did the authors also investigate the virulome and resistome at D30?

Author Response

  • Abstract

Line 28-30: Our data confirmed that antibiotic exposure modified bacterial pathogenicity and, in this situation, linezolid seems to be the best adapted therapeutic option to limit the emergence of resistant bacteria

Spread of linezolid-resistant S. epidermidis isolates (cfr-carrying plasmid) is well documented in the hospital setting, also with interspecies transfer to S. aureus. https://pubmed.ncbi.nlm.nih.gov/35425984/ In light of this, the assessment of linezolid superiority should be limited to the results of the current study

We modified the new version of the manuscript by moderating this sentence.

  • Results

-Line 63-64: is it possible to in specify the period of time that the isolates were obtained? I would also be interesting, and maybe also important for the interpretation of the results, any antimicrobial treatment given to the cases prior/during the time of detection of isolates in blood culture. And also confirm if the S. epidermidis isolates were obtained from 36 different patients.

The period of time was one complete year (1st January to 31st December). The 36 S. epidermidis used in this study were indeed obtained from 36 different patients. Only the strains coming from catheter and bloodstream (collected simultaneously) could be recovered. We have clarified this information in the “Material and methods” section.

Unfortunately, we did not have the necessary authorizations to access the patient data, including the antibiotics administered at the time of strains collection.

-Line 121-122: After 15 days of antibiotics exposure, all strains were resistant to vancomycin (MICs= 4mg/L), 3/5 to daptomycin (MICs= 2mg/L) and 2/5 to ceftobiprole (MICs= 122 4 mg/L).

According to EUCAST Clinical Breakpoint Tables v. 12.0, valid from 2022-01-01, Vancomycin MIC breakpoints for coagulase-negative staphylococci is S ≤ 4 mg/L and R > 4 mg/L. The authors interpret the breakpoints as if R ≥4 mg/ml for Vancomycin. This discrepancy has to be addressed as it has major implications for the interpretation of the results. In addition, ceftaroline MICs for coagulase-negative staphylococci is missing. Please comment.

We thank the reviewer for bringing this to our attention. This was incorrectly written in the previous version of the manuscript. We used the recommendations of the CA-SFM (Société Française de Microbiologie) (2021, V1.0). We apologize for this error and have rectified this in the new version of manuscript. Indeed, according to CA-SFM vancomycin susceptibility for coagulase-negative staphylococci is fixed to 2 mg/ml, hence our conclusions.

-Line 127: * S, susceptible; R, resistant according to EUCAST recommendations [27].

This may be wrong? Reference 27 is currently Dai, Y.; Wang, Y.; Liu, Q.; Gao, Q.; Lu, H.; Meng, H.; Qin, J.; Hu, M.; Li, M. A novel ESAT-6 secretion system-secreted protein 518 EsxX of community-Associated Staphylococcus aureus lineage ST398 contributes to immune evasion and virulence. Front Micro-519 biol 2017, 8, 819.

Thank you for highlighting this mistake. We have changed the reference in the new version of the manuscript

  • Materials and methods 

-Line 353-355: Antimicrobial susceptibility testing of isolates was performed by disc diffusion test 353 and broth microdilution method on Muller Hinton according to EUCAST recommendations. Please include which version (v 12.0?)

Version of CA-SFM recommendations (2021-V1.0) was added in the manuscript

-Line 181-184: It would be interesting to include Table 2 WGS data for the 15 sequenced strains. I suppose that this refers to train 5, 26, 82, 97 and 105 at D0, D15 and D30? 

Table 2 presents general genomic characteristics, virulome and resistome for initial strains (5, 26, 82, 97 and 105) at D0 only. Table 3 presents the SNP and so the genomic differences between D0 and D15. General characteristics/ virulome and resistome were identical. At D0 and D15 only SNP shown in table 3 highlighted a difference (mutation or deletion in mentioned genes) between strains that could be responsible for the antibiotic resistance. We differentiated these 2 tables to avoid confusion.

Did the authors also investigate the virulome and resistome at D30?

We also investigated general genomic characteristics, virulome and resistome at D30 but as no single SNP was found between D15 and D30 we did not present the data. A sentence was added in the manuscript to describe it.

Reviewer 2 Report

In this work, authors studied the effect of prolonged exposure (15 days) to sub-inhibitory concentrations of antibiotic (ceftobiprole, daptomycin, linezolid, vancomycin) on Staphylococcus epidermidis pathogenicity/virulence, which include enhanced antibiotic resistance, biofilm formation ability and expression of genes related with resistance and biofilm phenotype.

They demonstrated and concluded that for some S. epidermidis isolates, antibiotic exposure increased their biofilm formation capability, changed de gene expression profile and led to cross resistance between vancomycin and daptomycin.

Overall, this is an interesting study, which highlight the impact of antibiotic exposure on the “fitness” of S. epidermidis strains obtained from catheter-related bacteremia. Another key point of this work is that it highlighted that care should be taken when applying antibiotic treatment, as this can further expand selective pressure, resistance and biofilm formation capability. Besides, it reinforces that therapeutic options should be revised, particularly when biofilm infection are involved. However, these statements are not new.

Although of the contribution that results from this study could provide to community in general and specially to medical one, there are some gaps/limitations that should be considered. I appoint some suggestions:

- In general the English language could be improved.

- In introduction, more relevance to biofilm problematic should be included.

- Please revise “…showed a majority of isolates…” (pp. 2, line 65).

- Please revise “…was used as nutrient for C. elegans served as a negative control” (pp. 2, line 71).

- Bacterial isolates was obtained from “University Hospital of Montpellier” or “Nîmes University Hospital” (pp. 2, line 64; pp. 11, line 345).

- It would be better “biofilm formation potential” instead of only “biofilm potential (BP)”. Please revise in all manuscript (pp. 3, line 90).

- “All vancomycin-resistant S. epidermidis harbored…”, This is true for all vancomycin-resistant S. epidermidis?? (pp. 5, line 130).

- It would be better “selective pressure”, instead of “selection pressure”.

- Please revise “…induce resistant isolates…” (pp. 9, line 257).

- Please explain/be more precise “…corroborating a previous work” (pp. 10, line 291).

- Be more clearer “…were noted, confirming that the strains were the same according to Ankrum 323 and Hall criteria)” (pp. 10, line 323).

- “…involved in biofilm” formation??

- In general the “Material and Methods”, should be more described, particularly “antibiotic exposure methodology”. For instance, for this method was used solid or liquid medium during antibiotic exposure?

Also, this is not the first time that these methods were applied. Please provide the respective references, for each method presented.

- “…standardized  bacterial cultures”, please specify (pp. 11, line 380).

- What exactly does “partial biofilm minimal inhibitory concentration”? 50%?? Be more precise.

- One of the main concerns, is why the biofilm studies was conducted with early biofilms “4 hours”, and not with more mature ones (e.g. 24 hours)? at 4 o'clock practically only have adhesion to the surface…

 - It would be great, if the results were more corroborated and discussed.

- References should be more up-to-date.

Considering, all that was mentioned above, I think that the submitted manuscript should be careful revised and improved, before be accepted.

Author Response

- In general the English language could be improved.

A native English medical writer has reviewed this version of the paper.

- In introduction, more relevance to biofilm problematic should be included.

As suggested, we have highlighted biofilm problematic on medical device in introduction.

- Please revise “…showed a majority of isolates…” (pp. 2, line 65).

This sentence was changed

- Please revise “…was used as nutrient for C. elegans served as a negative control” (pp. 2, line 71).

This sentence was modified.

- Bacterial isolates was obtained from “University Hospital of Montpellier” or “Nîmes University Hospital” (pp. 2, line 64; pp. 11, line 345).

We have specified that bacterial isolates were obtained from Montpellier University Hospital (in Department of Infectious Diseases, Pr Jacques Reynes)

- It would be better “biofilm formation potential” instead of only “biofilm potential (BP)”. Please revise in all manuscript (pp. 3, line 90).

This revision was done according to the comment

- “All vancomycin-resistant S. epidermidis harbored…”, This is true for all vancomycin-resistant S. epidermidis?? (pp. 5, line 130).

We have modified this sentence for clarity.

- It would be better “selective pressure”, instead of “selection pressure”.

“Selection pressure” was changed for “selective pressure”

- Please revise “…induce resistant isolates…” (pp. 9, line 257).

We have revised the sentence.

- Please explain/be more precise “…corroborating a previous work” (pp. 10, line 291).

We modified this paragraph and better described data obtained in the previous work.

- Be more clearer “…were noted, confirming that the strains were the same according to Ankrum 323 and Hall criteria)” (pp. 10, line 323).

We added information to explain the criteria Ankrum and Hall criteria.

- “…involved in biofilm” formation??

We changed this sentence according to the comment.

- In general the “Material and Methods”, should be more described, particularly “antibiotic exposure methodology”. For instance, for this method was used solid or liquid medium during antibiotic exposure?

We completed the “Material and methods” section to be more precise and in particular we added description for the antibiotic exposure experiments.

Also, this is not the first time that these methods were applied. Please provide the respective references, for each method presented.

References for each method used in this study can be found in the corresponding paragraph.

For instance, “Partial MICb were measured using Antibiofilmogram® test (Biofilm Control) according to the manufacturer’s recommendations [58]”

- “…standardized  bacterial cultures”, please specify (pp. 11, line 380).

We added information on the standardized bacterial cultures.

- What exactly does “partial biofilm minimal inhibitory concentration”? 50%?? Be more precise.

We have defined the biofilm inhibitory concentration.

- One of the main concerns, is why the biofilm studies was conducted with early biofilms “4 hours”, and not with more mature ones (e.g. 24 hours)? at 4 o'clock practically only have adhesion to the surface… 

The Biofilm Ring Test (BRT) gives information about early stages of biofilm formation and in particular the first stage, the adhesion. The method can be used to screen strains able to induce a rapid biofilm formation and is commonly used for different species. (see Olivares E, et al. Clinical Biofilm Ring Test® Reveals the Potential Role of β-Lactams in the Induction of Biofilm Formation by P. aeruginosa in Cystic Fibrosis Patients. Pathogens. 2020 Dec 19;9(12):1065. doi: 10.3390/pathogens9121065. Di Domenico EG, et al. Development of an in vitro Assay, Based on the BioFilm Ring Test®, for Rapid Profiling of Biofilm-Growing Bacteria. Front Microbiol. 2016 Sep 21;7:1429. doi: 10.3389/fmicb.2016.01429).

In addition, some of our previous studies have shown a correlation between BRT and the ability of the strains to ultimately form a mature biofilm (Pouget C, et al. A Relevant Wound-Like in vitro Media to Study Bacterial Cooperation and Biofilm in Chronic Wounds. Front Microbiol. 2022 Apr 6;13:705479. doi: 10.3389/fmicb.2022.705479.).

The time of four hours was established following our experience of this test. This time allows to separate very early, early and non biofilm formers in S. aureus strains.

Finally, the BRT is a commercialized and highly reproducible test that is not the case with the conventional crystal violet test (see for example Amador CI et al., High-throughput screening alternative to crystal violet biofilm assay combining fluorescence quantification and imaging. J Microb Meth 2021, 190, 106343.; Kragh KN et al. Into the wel-a close look at the complex structure of a microtiter biofilm and the crystal violet assay. Biofilm. 2019, 1, 100006).

 - It would be great, if the results were more corroborated and discussed.

The discussion part has been modified in the new version of the manuscript.

- References should be more up-to-date.

We added recent references to update the discussion.

Reviewer 3 Report

The study is well executed and i recommend it for publication. one thing I can see most of the references are old (more than five years), authors need to update them as we can find recent studies relevant to the study.

Author Response

-The study is well executed and i recommend it for publication. one thing I can see most of the references are old (more than five years), authors need to update them as we can find recent studies relevant to the study.

We added some recent references in the discussion part.

Round 2

Reviewer 2 Report

I am satisfied with the corrections and response to my comments.